# Improving health care from the bottom up: Factors for the successful implementation of kaizen in acute care hospitals

Kosta Shatrov [1,2] *, Camilla Pessina [1], Kaspar Huber [3], Bernhard Thomet [4], Andreas Gutzeit [5,6,7], Carl Rudolf Blankart [1,2]

**1** KPM Center for Public Management, University of Bern, Bern, Switzerland, **2** Swiss Institute for Translational and Entrepreneurial Medicine, Bern, Switzerland, **3** Hirslanden Klinik St. Anna, Lucerne, Switzerland, **4** Corporate Office, Hirslanden AG, Glattpark, Switzerland, **5** Department of Radiology, Paracelsus Medical University, Salzburg, Austria, **6** Department of Chemistry and Applied Biosciences, Institute of Pharmaceutical Sciences, ETH Zurich, Zurich, Switzerland, **7** Institute of Radiology and Nuclear Medicine and Breast Center St. Anna, Hirslanden Klinik St. Anna, Lucerne, Switzerland

* kosta.shatrov@unibe.ch

## Abstract

### Background

Kaizen—a management technique increasingly employed in health care—enables employees, regardless of their hierarchy level, to contribute to the improvement of their organization. The approach puts special emphasis on frontline employees because it represents one of their main opportunities to participate directly in decision making. In this study, we aimed to (1) understand the experiences of nurses in two hospitals that had recently implemented kaizen, and (2) identify factors affecting the implementation of the technique.

### Methods

By means of purposeful sampling, we selected 30 nurses from different units in two private acute care hospitals in Switzerland in May 2018. We used the Organizational Transformation Model to conduct semi-structured interviews and perform qualitative content analysis. Lastly, originating from Herzberg's motivation theory, we suggest two types of factor influencing the implementation of kaizen—hygiene factors that may prevent nurses from getting demotivated, and motivational factors that may boost their motivation.

### Results

Nurses generally experienced kaizen as a positive practice that enabled them to discuss work-related activities in a more comprehensive manner. In some cases, however, a lack of visible improvement in the workplace lowered nurses' motivation to make suggestions. Nurses' attitudes towards kaizen differed across both hospitals depending on the available managerial support, resources such as infrastructure and staffing levels.

**Data Availability Statement:** All relevant data are within the manuscript and its Supporting Information files.

**Funding:** The authors received no specific funding for this work. The authors KH, BT, AG are employees of Hirslanden AG, the owner of the two acute care hospitals investigated in the study. The funder provided support in the form of salaries for authors KH, BT, AG, but did not have any role in the study design, data collection and analysis, decision to publish, or preparation of the manuscript. The specific roles of these authors are articulated in the 'author contributions' section.

**Competing interests:** All authors of this manuscript have read the journal's policy and have the following competing interests: KS, CP and RB declare that they have no conflicts of interest. KH, BT, and AG are employed by Hislanden AG, the hospital organization that was examined in this study. The commercial affiliation of the authors KH, BT, and AG does not alter our adherence to PLOS ONE policies on sharing data and materials. Hislanden AG does not profit financially or otherwise from the study results. All authors declare that no other competing interests exist.

## Conclusions

From our findings, we derived several coping strategies to help health practitioners implement kaizen for the benefit of their organization and employees: Strong managerial support, appropriate use of kaizen tools, and a greater sense of team cohesion, among other factors, can influence how effectively hospital teams implement kaizen. To reap the benefits of kaizen, hospital managers should promote the exchange of opinions across hierarchy levels, allocate the necessary resources in terms of personnel and infrastructure, and show nurses how the technique can help them improve their workplace.

## Introduction

Kaizen is a management approach that aims for the continuous, incremental improvement of an organization. If implemented successfully, it empowers employees, regardless of their hierarchy level, to address problems and take actions to solve them [1]. The concept promotes organizational change and a culture of continuous improvement with the ultimate goals of avoiding waste and increasing quality throughout the organization [2]. In this context, improvement is regarded as a recurring process and not a project with a predefined timeframe. Kaizen encompasses the real-time assessment and quick implementation of ideas from the bottom up, ultimately resulting in small but substantial improvements [3]. Employees are able to make suggestions and decide which ones to implement [4], which typically requires infrastructure that reduces barriers to reporting problems, and facilitates the adoption of ideas. Examples include suggestion boxes, discussion rounds, and interactive dashboards tracking how ideas are implemented.

Increased health spending in recent decades has led many health care providers to take measures to contain costs. This trend is especially true for countries that refund inpatient care treatments through diagnosis-related groups, a reimbursement system that induces hospitals to focus on cost-efficiency by providing financial incentives to deliver health care at below-average costs. When faced with the objective of reducing costs and improving quality, hospitals that organize their activities around core business processes in a structured way obtain higher levels of efficiency [5]. Kaizen offers managers the opportunity to provide high-quality care and increase organizational performance while maintaining cost efficiency. Promoting employee engagement affects performance at the organizational, team, and individual levels [6], and also boosts job satisfaction and commitment to the workplace [7]. By expanding the role of less senior employees, whether formally or informally, managers can reinforce employee participation in decision making [7]. More generally, continuous improvement techniques result in a series of tangible outcomes, such as reductions in errors or costs, and intangible outcomes, such as increased autonomy and employee motivation [8, 9]. Given that continuous improvement techniques like kaizen can benefit both the organization (e.g., by reducing costs), and the individual (e.g., by increasing job satisfaction) [3, 9, 10], it is hardly surprising that they, as well as the broader concept of lean methodology, have been applied widely by health care organizations in recent decades [11–14]. Encouraging employees to contribute to decision making has thus become a widespread management practice, which, however, varies widely in terms of its implementation and outcomes.

While research on continuous improvement techniques in health care is plentiful, it has given short shrift to cases of less successful implementation [11], and to the interrelationship

between implementation and staff [12]. This gap in the literature is unfortunate considering that (1) much can be learned from failures of implementation [11, 15] and (2) integrating frontline employees is crucial to the success of continuous improvement techniques [16, 17]. We aimed to address this gap by exploring the implementation of kaizen in two private acute care hospitals in Switzerland. In doing so, we capitalized on three aspects of our study design to advance scholarly and practitioner knowledge of continuous improvement management techniques: First, we exploited the variation between two distinct approaches to implementing kaizen, e.g., in terms of how managers set goals aiming to measure the implementation of the technique. Second, we sought to understand the experiences of nursing staff with the implementation of kaizen. We chose to focus on this professional group because both sites had implemented kaizen specifically to provide nurses with the opportunity to improve their workplace. Additionally, nurses—unlike physicians—had few other opportunities to contribute to decision making related to the organization of the hospitals of interest. Lastly, by offering evidence from outside the widely studied cases of the United States and the United Kingdom [11, 12], we aimed to support mutual learning among countries with different health systems and traditions of nursing education and practice.

## A general comparison of public and private hospitals in Switzerland

Inpatient care in Switzerland is reimbursed by health insurers at a flat rate based on diagnosis related groups [18]. This system was introduced in Switzerland in 2012 and applies to all hospitals regardless of their ownership status or profit-orientation. In 2013, about 30% of all acute care hospitals in Switzerland were privately owned, and provided mostly standard surgical and elective treatments [18]. Farsi and Filippini have shown that there are generally no significant differences in cost-efficiency between Swiss hospitals of different ownership status; rather, the cost-driving factors are higher levels of teaching activities in some hospitals and a broader range of specialization in others, and both are associated with lower cost-efficiency [19].

Although the two hospitals that participated in our study were private and profit-oriented, they were comparable to public and non-profit hospitals in terms of a wide range of indicators, including the number of outpatient consultations, number of days in inpatient care, staffing (measured in full-time equivalents), and the overall complexity of treatments provided (measured by a case mix index) [20]. However, there are some important differences between the hospitals examined in this study and the average Swiss hospital, public ones in particular. The most important difference is that the majority of specialist physicians in the two participating hospitals were not employed by the hospitals but affiliated with them and hired as locums. These physicians provided treatment to patients by renting the infrastructure of the hospital and a team of supporting physicians (anesthetists, emergency and intensive care physicians). Some of the specialists were nevertheless directly employed by the hospitals or had post-doctoral degrees and a range of teaching tasks, something that would be more typical of a university hospital. Second, compared to the Swiss median, the two hospitals treated a larger proportion of patients who had a private insurance plan. Such plans enable the patients to choose their treating physician and to stay in a single-bed room. Lastly, although both public and private hospitals have the same range of tasks and obligations in Switzerland, private hospitals are not eligible for public funding in certain domains, such as investments in building infrastructure.

Importantly, the two hospitals are not exclusive private clinics, but an integral part of the Swiss health system in their provision of general acute care. In Switzerland, private hospitals are reimbursed in the same manner as public ones, resulting in financial incentives that are independent of ownership status and profit orientation. The Swiss hospital sector comprises a

spectrum ranging from non-profit university hospitals funded by public funds that usually treat the most complex cases to niche private hospitals working almost exclusively with privately insured patients and focusing on selective treatments. Overall, the two hospitals we examined are—irrespective of the important features discussed above—fairly similar to what can be described as the average Swiss hospital.

## Methodology

### Ethics approval

The study design, the interview guide, and our data management concept were approved by the ethics committee of the Faculty of Business, Economics and Social Sciences of the University of Bern (Date: 2018-05-08; Process number: 180503_1).

### Study design

The main purposes of our study design were to (1) explore nurses' perceptions of kaizen, and (2) derive factors that affect the implementation of the technique from a direct comparison of two acute care hospitals. Based on the Organizational Transformation Model (OTM) developed by Lukas, et al. [21], we created a semi-structured interview guide (the full version of which can be found in the S1 Table). Table 1 provides an overview of the content and structure of the interview guide.

We would expect that OTM is a suitable framework for our study design because it was developed during a series of case studies in private sector health care organizations. The OTM describes five drivers that are crucial to transforming patient care successfully: (1) impetus to transform; (2) leadership commitment to quality and change; (3) staff engagement in improvement initiatives; (4) alignment of an organization's goals with resource allocation; and (5) overcoming of boundaries between the constituent parts of the organization so that it operates as a fully interconnected system, pursuing the overarching goals of the organization.

In order to gain insights that would be valuable to both scholars and practitioners [22] and to extend the scope of qualitative research in this area [23], we added an additional driver to the OTM framework—employee commitment—to describe employees' attachment to the

**Table 1. Sample interview questions.**

| Organizational transformation via kaizen | |
| --- | --- |
| Impetus to transform | • In your opinion, why were you given the opportunity to make suggestions for improvement with the kaizen technique?<br>• How important do you think is it to your supervisor that you work independently? |
| Leadership | • How would you describe the general support of hospital management in your daily job?<br>• What was the reaction of your supervisor to a kaizen suggestion you made? |
| Staff engagement | • Have you already made any suggestions?<br>• Could you give an example of a suggestion you have made for solving a specific problem? Would your participation behavior change if you had no access to the kaizen tools? |
| Alignment | • What was the impact of your suggestions? How were they implemented?<br>• Do you have specific responsibilities within the kaizen technique and how do you exercise them? |
| Integration | • The staff in your ward is encouraged to report problems and make suggestions for improvement, is that correct? Can you please describe what this looks like in practice?<br>• What is the impact of kaizen on the way your ward operates?<br>• How important is the opinion of the nursing staff at the hospital? |
| Commitment | • Does performing kaizen affect your attitude towards the hospital as employer?<br>• Does the opportunity to express your opinion have any influence on your willingness to continue to work at the hospital? |

organization or to parts of it, such as their supervisors [24]. Organizational change initiatives may affect employee commitment to the change itself or to the whole organization [25]. Given the positive general association between commitment and participation [6, 26, 27], the opportunity to contribute to the improvement of the workplace may increase nurses' willingness to stay at a hospital.

## Data collection

Following the guidelines of Gill, et al. [28], we conducted semi-structured interviews with 30 nurses in May 2018. The two hospitals (hereinafter referred to as Hospital A and Hospital B) had recently implemented kaizen and were open to the idea of exploring nurses' experiences with the implementation of the technique. We selected these two hospitals because medical experts and representatives of their management teams (AG, KH, BT) had indicated that the sites were fairly similar in terms of size and specialization, but differed in the way kaizen had been applied. Both hospitals were medium in size, profit-oriented, located in Switzerland, and belonged to the same corporation. The hospitals were comparable in terms of the average length of stay (5.0 vs. 5.6 days), bed occupancy rate (84.8% vs. 84.2%), number of newborns (827 vs. 860), and number of emergency admissions (4,316 vs. 4,212) in the fiscal year 2017/2018. In the same fiscal year, the first hospital (Hospital A) was smaller than the second one (Hospital B) in terms of the number of beds (196 vs. 333), patients (12,198 vs. 18,389), and employees (1,377 vs. 2,128) [29].

In Hospital A, an implementation working group, whose main objective was to set out the conditions for the nursing teams to be able to start contributing ideas, was appointed by the management team of the hospital at the beginning of the implementation phase. To communicate the concept of kaizen to the staff in all units, the working groups defined a set of general and specific aims. While the general aims involved encouraging nurses to seek to identify problems in the workplace and subsequently solve them, the specific ones defined concrete measures intending to facilitate collaboration. Such specific measures included the advice that kaizen meetings should be held on a regular basis and that nurses should aspire to be role models to their colleagues from other units by working with and for each other. Lastly, the process of implementing kaizen was represented graphically in the recreation rooms of Hospital A and defined in five steps as follows: (1) identify waste; (2) make an improvement suggestion; (3) prioritize the suggestion and define action to implement it; (4) take the defined actions; (5) measure the success of the actions.

The initial phase of implementing kaizen was defined in a similar way in Hospital B. Together, the quality management team and the nursing team of the hospital elaborated a strategy to design the way in which kaizen would be implemented in all inpatient care units. The strategy of Hospital B sought to encourage employees to question existing working procedures, inform their colleagues in case action is required, and eventually make a collective effort to improve their workplace. In addition, responsible persons received special training and subsequently served as an important reference point to the rest of team.

By means of purposeful sampling [30], we selected 15 nurses from each hospital to obtain an information-rich sample that was balanced in terms of age, gender, tenure, and specialization. Our sample included nurses from several units, including orthopedics, gynecology, and thyroid diseases treatment. The nurses comprised 28 women and two men, of whom 11 were in training—some of whom had recently started working for the hospital—and 19 had professional qualifications. Five of the latter occupied senior positions, such as head nurse. We informed all nurses in advance that they would receive a small thank you gift after completing the study. After being introduced to the study purpose and design, all participants gave their

written informed consent to be interviewed and recorded. Interviews lasted an average of 26 minutes [range: 15–38 min].

## Analysis

The interviewer (CP) used the recordings to (1) transcribe the interviews, including pauses, filler words (e.g., "hmm"), and important non-verbal reactions (e.g., laughing), and, (2) to conduct a qualitative content analysis of the transcribed interviews in MAXQDA (Version 18.0.8). First, we identified patterns in the text and grouped these into overarching themes to develop an in-depth understanding of the interviews. The full list of themes, categories, and marker words we used to conduct the content analysis of the transcribed interviews can be found in S2 Table. The next step was to summarize the interviews by exporting quotes into a structured matrix organized by site, interviewee, and question (see S3 Table). Then, we applied the OTM framework revising the influencing factors suggested by Sullivan, et al. [31], who recently validated the framework by assessing the implementation of a pilot health care services program in long-term care facilities. In the next step, to enhance the credibility of analysis [32], two researchers (KS and RB) used the OTM framework to analyze the interviews. The two analysts independently assigned each of the quotes to one of the six influencing factors and their sub-categories: (1) nurses' perspectives, (2) job commitment and satisfaction, (3) team dynamics and processes, (4) infrastructure availability and adoption, (5) human resources and staffing, and (6) resource allocation and culture. Auxiliary evidence from the interview notes was considered if needed. After an initial agreement of 68% between the two analysts, we refined some sub-categories and developed categorization rules to avoid ambiguity (see S1 Fig for illustration and S4 Table for an overview of the categorization rules). In a second iteration, applying the categorization rules increased overall agreement between authors to 90%. In the final iteration, the two analysts resolved any disagreements in their categorization by reaching consensus through discussion.

After analyzing the interviews using the OTM framework, we interpreted our findings within the scope of the two-factor motivation theory, which posits that certain factors influence employee motivation at the workplace [33]. The theory distinguishes between hygiene factors that lead to dissatisfaction and motivational factors that boost employee satisfaction. The core of the theory is that both types of factor do not build a continuum–that is, hygiene factors cannot yield satisfaction, and motivational factors are not associated with dissatisfaction. More specifically, hygiene factors comprise the working conditions that managers need to provide to prevent their employees from losing motivation and becoming unhappy with their workplace (e.g., due to unsafe work practices or conditions), whereas motivational factors comprise the working conditions that should increase employees' job satisfaction and keep their motivation high (e.g., work practices promoting employees' sense of achievement). Even though the validity of Herzberg's theory has been called into question because working conditions have changed significantly since the theory was initially proposed in the late 1950s, Bassett-Jones and Lloyd have demonstrated that the underlying idea of the two-factor theory still has utility in a contemporary organizational context [34]. By interpreting our findings within the scope of the two-factor motivation theory, we aim to provide health care practitioners with insights into factors—at the individual and organizational levels—that may either increase or decrease the motivation of nurses to participate in kaizen.

## Results

Based on our examination of how hospital nurses used kaizen, we identified a range of specific individual and organizational factors that affected the implementation of the management

**Table 2. Illustrative quotes assigned to corresponding influencing factors.**

| Influencing factor | Sub-category | Illustrative quote |
|---|---|---|
| Nurses' perspectives | • Nurses participate in kaizen by making suggestions and/or implementing ideas<br>• Nurses support the use of kaizen at the hospital<br>• Other perceptions of kaizen and its importance | "It's hard to say. . . [what the importance of kaizen is.] Well, it is important, but yes, of course, I also have more important tasks." (Hospital A/08)<br>"Our ideas mostly refer to improvements in terms of quality and time management. For example, how we can organize rooms to save space, which additional equipment we need or don't use so often." (Hospital B/20) |
| Job commitment and satisfaction | • Kaizen increases commitment to the hospital<br>• Kaizen increases overall job satisfaction | "I enjoy sharing my opinion and making suggestions, but. . . I'd still work here even if there were no kaizen." (Hospital A/14)<br>"No. No [link between kaizen willingness to stay]. We are free people, we can quit and go or we can stay." (Hospital B/29) |
| Team dynamics and processes | • How well staff fit with kaizen either through commitment to the program vision and/or experience and skills needed for successful implementation<br>• Teamwork, coordination, and cohesion in terms of how well hospital staff work together, support each other, and create a collaborative work environment<br>• There is a clearly pre-defined process/structured way in which kaizen works | "The head nurse collects the opinions of everyone, and we discuss our ideas. She considers our suggestions and draws conclusions from our opinions. It works out very well." (Hospital A/03)<br>"If many of my colleagues come up with good suggestions, then that motivates me to think more about what can be improved." (Hospital B/18)<br>"The team meeting gives us structure [. . .] At the same time, it's hard to make a meaningful contribution if you desperately need to do something or you're quite busy." (Hospital B/21) |
| Infrastructure availability and adoption | • Infrastructure needed for the sustainable implementation of kaizen is available and accessible, e.g. dashboards, regular meetings<br>• Nurses make use of the kaizen infrastructure provided | "I like the circle [a pie chart illustrating the progress of implementing suggestions; a part of the kaizen dashboard] [. . .] it's like a reminder." (Hospital A/06)<br>"We have a kaizen training, in which the system is explained and applied. I think this is good for employees who are new and have no previous experience with the system." (Hospital A/07) |
| Human resources and staffing | • Constraints with existing staff, e.g. short-staffed, not enough time to include kaizen in the work routine<br>• High staff turnover/extensive use of agency staff | "What I find very often a pity, quite a pity, is that the nursing staff in general has far too little time for nursing." (Hospital A/09)<br>"We are understaffed and rely on agency staff. These employees don't belong to us. [. . .] They don't have the same responsibilities as we do. And if there's something they don't know, we have to spend extra time to help them." (Hospital B/29) |
| Resource allocation and culture | • The adequacy of resources dedicated to kaizen implementation and/or achieving sustainable results<br>• Management shows general support and/or is persistent in encouraging the implementation of kaizen<br>• Management has established a culture that promotes open dialog and/or tolerates failure | "No, that's no problem [if colleagues don't agree with me]. It's really a platform where everybody can express their own opinion." (Hospital A/04)<br>"Sometimes we work with very dominant physicians, and you can feel the hierarchy. In some situations, I thought 'I'd better not say anything' [. . .] I lacked the courage to speak up, because of the hierarchy." (Hospital A/14)<br>"It's difficult, when you address a problem, and your opinion is kind of accepted, but it's always accompanied by an excuse that defends the underlying problem. That makes it a bit difficult to discuss in the first place." (Hospital B/27) |

approach. Overall, nurses experienced kaizen as a positive practice that promoted teamwork and provided them with an opportunity to participate in decision making and contribute to the continuous improvement of the hospital. Most nurses in both hospitals participated in kaizen by attending regular meetings, reporting problems, and making suggestions regarding the availability of resources and patient well-being. Table 2 presents some illustrative quotes assigned to the respective influencing factor derived from the modified OTM framework. Additional illustrative quotes—originating from nurses of different age, gender, tenure, and specialty—that support our findings have been included in the results section.

The remainder of this section is organized around the six influencing factors derived from the OTM, starting with the individual factors and then moving on to the organizational ones.

### Individual factors

**Nurses' perspectives.** Experience and seniority seemed to affect the willingness of nurses to share their opinion in front of colleagues, especially in Hospital A:

*'Four and a half years ago, when I was still a trainee, I had the impression that we weren't allowed to say very much, and I was a bit more cautious about what I was allowed to say and what not; and now, as a full-time employee, I've noticed that every opinion counts' (Hospital A/07)*

Less experienced nurses were indeed often reluctant to speak their mind. Nonetheless, they appreciated that kaizen gave them an equal chance to contribute ideas. In Hospital B, however, two nurses stated that they had had no experience with kaizen in their hospital so far, and three others suggested that they had not used kaizen in their unit (although they went on to describe having used structures typical of kaizen, such as dashboards, regular team meetings, and goal setting).

**Job satisfaction and commitment.** At both hospitals, kaizen seemed to evoke positive feelings among many nurses, who reported feeling valued, understood, and confident. The majority of our interviewees appreciated having had the chance to contribute their ideas, and some nurses noted that working as a team towards the goal of improving working conditions had increased their job satisfaction somewhat. In addition, many nurses—mostly in Hospital A—agreed that kaizen promoted individual decisional power. Although not decisive in itself, the kaizen-related policy that every person's opinion counts increased nurses' willingness to work for the hospital, albeit only marginally and only in Hospital A. Especially in that hospital, the feeling that one's ideas were being considered appeared to boost overall motivation. In contrast, some nurses in Hospital B felt that the new management practice was not being implemented properly—partially because of the profit-orientation of the hospital—generating a negative emotional response and, to a certain extent, decreasing employee motivation:

*'I would like to be able to provide my patients with high-quality, evidence-based care, and that's only possible if certain preconditions are met. If [. . .] you are constantly short on staff because of the profit situation here in a private hospital [. . .] then you get demotivated' (Hospital B/27)*

### Organizational factors

**Team dynamics and processes.** Nurses in both hospitals generally agreed that team meetings promoted collaboration:

*'Once a month, the kaizen [meeting] takes place and the employees meet in the office. [. . .] The problems are then discussed within the team, and we see what can be improved, what the options are, and which person or people are responsible for implementing it' (Hospital A/12)*

Moreover, discussing work-related problems made nurses feel part of the team and the hospital. Many nurses said that they appreciated the contributions of new employees because they felt their perspectives were innovative and unconventional. The cross-hierarchy exchange, moreover, was generally regarded as meaningful and constructive. However, nurses also indicated that there was still a need for better coordination of working routines, including agenda-setting:

*'Well, sometimes there are too many [kaizen] targets, and you don't even look at them anymore. I think that's a bit of a shame' (Hospital B/30)*

Both hospitals implemented kaizen in a similar way. Regular discussion rounds were introduced—up to once in a fortnight in Hospital A and on a weekly basis in Hospital B—and kaizen dashboards were installed in all units. The dashboards represented whiteboards, so nurses were able to make improvement suggestions and track their status by documenting their ideas using markers and post-it notes. The dashboards had been placed in easily accessible places such as break rooms and kitchens. Nevertheless, there were also some differences. Hospital A started implementing kaizen in 2010; Hospital B did so in 2011—first in all inpatient departments, and two years later in the intensive care unit. Although the dashboards in both hospitals were divided into sections that were devoted to the tasks of contributing ideas and defining a set of actions for their implementation, Hospital B did not use a pie diagram to visualize the implementation status of ideas. Hospital A offered compulsory introductory training to all nurses. In Hospital B, however, the quality management team visited units to answer any questions nurses had during the initial phase of implementation. Hospital A set the goal of implementing 20 ideas in each unit per year, whereas Hospital B aimed to have 36 meetings dedicated to kaizen during the first year of implementation. Within this year, Hospital A implemented 958 suggestions in total, whereas Hospital B implemented 321 suggestions.

In Hospital A, all nurses were encouraged to make suggestions, which were then evaluated and prioritized by an assigned person, and implemented by the entire team. Nurses in this hospital seemed aware that they had to contribute ideas to make kaizen work, and emphasized that the each-opinion-counts policy reinforced team cohesion. These perceptions were less evident in Hospital B, where a few employees nonetheless stated that they associated kaizen with meeting colleagues and discussing current issues.

**Infrastructure.** The majority of nurses highlighted that the regular discussion rounds were important because they animated everyone to exchange views and contribute ideas. Nurses at Hospital A appreciated tools such as dashboards and sticky notes, because they facilitated the flow of information and implementation of change initiatives. In the same hospital, one nurse asserted that kaizen tools encouraged less motivated employees to become more active. Dashboards, for instance, enabled ideas to be submitted and prioritized, tasks to be assigned, and their implementation status to be tracked:

> *'Sometimes there are things that can be implemented immediately [. . .] and sometimes there are things that need to be purchased first, and that takes longer. That's why we have [. . .] something like a cake. . . [a pie chart]. It has four parts, and you can always fill in the part as soon as the process has been completed' (Hospital A/04)*

Monitoring tools and introductory lessons, which were highly appreciated in Hospital A, were also present—although less common—in Hospital B:

> *'No, not really [in response to the question whether an introductory training had been offered]. I cannot really tell you [how kaizen works]' (Hospital B/18)*

In Hospital B, kaizen was occasionally regarded as a tool only to address problems rather than as opportunity to trigger change in a proactive way, and some nurses even felt that kaizen could be applied only if somebody else made a suggestion. Kaizen was sometimes described as time consuming because of the many meetings and occasional discussions of the correct way to implement the technique that were taking place. One nurse in Hospital B suggested, however, that kaizen structures and responsibilities had recently been defined more clearly.

**Human resources and staffing.** Most nurses agreed that nursing is a tough job and prioritizing tasks is demanding. Overall, nurses did not always manage to engage in kaizen because

they assigned higher priority to patient care. Nevertheless, nurses—mostly those in Hospital A —were aware that kaizen was an integral part of their work and even a way to reduce workload over the long term. Stress, however, was a recurring topic at both sites. Describing their daily routines, nurses often suggested that they felt under pressure. In Hospital B, stress levels seemed to be higher due to job fluctuation, the use of agency staff, and a lack of manpower:

*'I think time management is a huge problem, because creativity [. . .] takes time, and employees simply do not have time for that' (Hospital B/17)*

Many nurses who had regular contracts at Hospital B did not regard agency nurses as equal team members, nor did they support the policy of hiring agency staff, who they felt put in less effort and were less familiar with the working methods. Additionally, the nurses in this hospital occasionally attributed difficulties in implementing kaizen to the agency staff:

*'Well, we are currently using it [kaizen] a bit less because there is an extreme shortage of staff, and so many agency staff are coming in; and the agency staff don't participate in kaizen— they take care of their patients and that's it' (Hospital B/19)*

**Resource allocation and culture.** Organizational culture was another key factor that affected participation. In Hospital A, one nurse suggested that individual units probably interpreted kaizen differently according to their own culture:

*'And, naturally, depending on the culture [of the hospital department or individual stations], kaizen is implemented differently; because to keep it going with a high level of commitment you need a certain culture of openness on the team, so that you can sometimes also suggest an idea that might sound a little bit crazy—maybe something will come out of it' (Hospital A/10)*

Nurses in Hospital A seemed to be generally satisfied with the support they had received from hospital management and also cited the role of the head nurse as a leading figure. They also agreed more often than nurses in Hospital B that management was open to new ideas. Though more pronounced in Hospital B, hierarchy was present in both hospitals:

*'They [the physicians] don't say "hello", they don't look you in the eye. . . you often have the feeling they are something much better [. . .] Well, not all the doctors, but many are like that' (Hospital B/25)*

Nevertheless, nurses in both hospitals admitted that they needed more supervision and that they expected a person to be in charge of implementation (e.g., head nurse) and guide them in practicing kaizen.

Some nurses in Hospital B attributed lukewarm participation levels to a lack of sustainable results in their units:

*'At the beginning it [kaizen] has an effect–I would say for about [. . .] 4 weeks, or even only for 10 days, and then a lot, not everything, but a lot is forgotten' (Hospital B/28)*

In addition, the long time span between contributing an idea to improve the workplace and adopting it reinforced the view that kaizen was a rather laborious approach. This being said, many Hospital B nurses agreed that they needed to invest more time in kaizen to improve their work environment over the long run.

**Positive and negative cases in direct comparison**

Noting that the study of kaizen has focused so far on success stories, D'Andreamatteo, et al. [11] and Filser, et al. [12] advocate learning from examples of less effective implementation. With this in mind, we gained insights from comparing both of our participating hospitals in terms of (1) nurses' attitudes towards kaizen, (2) participative behavior, and (3) the results of implementing kaizen.

First, nurses' attitudes towards how kaizen had been implemented at the workplace differed between the hospitals. Nurses in Hospital B were less fond of engaging in kaizen, although they expressed their general willingness to contribute to the improvement of the hospital and recognized that kaizen is generally a useful practice. While Hospital B nurses reacted to the implementation of kaizen with a certain skepticism, nurses in Hospital A remained motivated to contribute ideas even if they did not always have time to adopt them immediately. Moreover, many nurses in Hospital B generally experienced kaizen as an additional workload imposed by management, whereas nurses in Hospital A were more likely to understand the approach as an integral part of the hospital's culture and their work that could help them improve working routines.

Next, the degree of participation in kaizen also differed. Even though kaizen structures were available in both hospitals, participation levels varied, for example due to a lack of leadership in individual units. Nurses who contributed their own ideas constituted the majority in Hospital A, but not in Hospital B. In the latter, it seemed that many nurses hardly ever made suggestions, though they still attended meetings and implemented kaizen projects.

Finally, the results of implementing kaizen differed between the hospitals. In Hospital A, most nurses agreed that kaizen improved their workplace. In contrast, some of the nurses in Hospital B indicated that kaizen did not lead to visible results at all times, which they often attributed to high levels of stress. Although nurses in Hospital A also agreed that nursing was stressful, they did not see a contradiction in taking part in kaizen alongside their other duties, and mentioned a sense of doing something meaningful when engaging in kaizen more often.

## Discussion

In this study of kaizen, we addressed the current research gap by focusing on the experiences of nursing staff and examining two opposing cases of kaizen implementation—one of which could be described as more successful than the other. We interviewed 30 nurses in two acute care hospitals in Switzerland. To obtain an information-rich sample, we selected nurses of different age, gender, tenure, and specialization. Our findings provide insights from a setting outside of the United States or the United Kingdom, which has been the almost exclusive focus of previous research in this area. In line with the literature [27], we found evidence that participation in decision making—through kaizen—may increase job satisfaction, albeit only to a limited extent.

Our main finding, however, is that there seem to be two types of factor that affect how kaizen is implemented in hospital care. In Table 3 we summarize our findings by assigning them to either of the two categories suggested in Herzberg's two-factor theory [33] in an attempt to specify which implementation measures affected nurses' motivation to participate in kaizen in what way—either by preventing them from getting demotivated, or by boosting their motivation. As in the results section, we distinguish in this summary between influencing factors at the individual and organizational levels. Building upon Herzberg's theory, we want to sensitize health care practitioners to the idea that there are certain working conditions they need to focus on in order to prevent nursing staff from losing motivation to participate in kaizen in the first place (hygiene factors), and that other working conditions may lead to nurses

**Table 3. Factors affecting nurses' motivation to participate in kaizen.**

|  | Hygiene factors | Motivational factors |
|---|---|---|
| **Individual level** | • Management and head nurse leadership and support<br>• Visibility of results | • Promoting everyday interactions<br>• Adopting employee suggestions<br>• Communicating the impact of kaizen activities |
| **Organizational level** | • Availability and accessibility of tools, e.g., dashboards<br>• Clearly defined processes and roles<br>• Team stability | • Culture of continuous improvement<br>• Team cohesion |

participating more intensively in the continuous improvement of their hospital organization (motivational factors).

## Strategies for the successful implementation of kaizen

Additionally, we identified several factors that influenced the implementation of kaizen in hospital care. Overall, the two hospitals we examined implemented kaizen by introducing a specific target to be achieved on a yearly basis, as well as regular discussion rounds and communication tools to facilitate the adoption of ideas. Hospital A seemed, however, to have capitalized on the potential of kaizen because it managed to implement the approach in a more structured and purposeful way, focusing on outcomes more than process. For instance, Hospital A defined the number of kaizen *ideas* to be adopted on a yearly basis as a target to be pursued by nurses, whereas Hospital B used the number of *meetings* held. Based on these and our other findings, we suggest six coping strategies for implementing the approach in hospital care. Depending on whether they relate to behavioral patterns that specific team members may need to pursue, or, instead, processes and structures that health professionals might wish to establish, the following strategies are assigned to either the individual or organizational level of acute care hospital systems:

**Individual level.** First, managers have to support nurses with expert assistance and advice. At both sites, nurses often expected managers not only to show them how a problem could be solved if they could not think of a solution right away, but to support them in implementing the solution. Many unexperienced nurses were reluctant to share their opinion, even though their ideas were often appreciated by their senior colleagues. Managers should therefore encourage the entire team to engage with kaizen and explain the benefits of sharing ideas. Some interviewees suggested that head nurses might also take on this role given that they are seen as important reference persons to other nurses. Indeed, support and leadership have been identified as essential preconditions for employee participation and the sustained implementation of change initiatives [3, 5, 11, 15]. Moreover, employees may have difficulty admitting that they have been doing things wrong for years and adjusting the way they work accordingly [5]. Our findings indicate that managers also have to create an open-minded work environment that promotes collaboration, self-criticism, and an efficient flow of information to make change possible. Authentic and trustworthy leadership may indeed improve the work environment, encourage team members to voice their concerns, and increase the perceived quality of care among nurses [35]. Conversely, a poor relationship with supervisors has been shown to decrease employees' willingness to contribute ideas [34].

Second, managers should promote everyday interactions across hierarchy levels and convincingly demonstrate to nurses that each of their opinions counts. Most nurses we interviewed enjoyed an increased sense of employee equality while participating in kaizen. Additionally, nurses widely associated kaizen with employee empowerment because it gave them a voice in decision making. This result is in line with previous research, which shows

that hospital employees appreciate being able to act more autonomously by participating in continuous improvement activities [36]. Although most nurses admitted that they generally had only limited scope to make managerial decisions, they enjoyed reorganizing their workplace through kaizen. We therefore conclude that nurses should not have the impression that supervisors make all the decisions. It is indeed important to leave some leeway for self-initiative and self-coordination [9] and to establish a corporate culture that eschews the traditional top-down approach to improvement initiatives [37, 38]. Additionally, giving employees more autonomy may boost motivation and augment the perceived value of their actions [39]. Leadership has the special task of finding the right balance between enabling the team to contribute and discuss ideas autonomously, and judiciously intervening in the prioritization and execution of suggestions.

Third, nurses need to see that their actions lead to meaningful results. In both hospitals, nurses seemed to lose patience and participate less if they had the feeling that kaizen was being implemented as an end in itself. In Hospital B, the absence of short-term results in some units was seen as proof that kaizen did not work, demotivating nurses. Therefore, managers should demonstrate to nurses that kaizen improves their workplace and the quality of care. Edmondson [15] as well as Mazzocato, et al. [4] also underlined the importance of continuously sharing insights and results with staff to keep motivation high. Indeed, the perceived success of adopted ideas has been shown to influence employee motivation to participate [39]. Our findings suggest, moreover, that motivation to participate may suffer if hospital staff sees kaizen only as a means to improve the financial performance of the hospital and not as a way to increase patient well-being. Furthermore, managers should recognize that nursing may be stressful, and nurses cannot always engage in kaizen because their main obligation is to care for patients. Nevertheless, to make kaizen work, nurses need to participate regularly.

**Organizational level.** Fourth, managers need to create a culture of continuous improvement. Many nurses had the impression that physicians did not always accept their suggestions or take their concerns seriously. This is unfortunate given that open dialog and a change-friendly work atmosphere have been shown to be essential for successful continuous improvement [36]. With this in mind, both physicians and managers should continuously encourage nurses to report problems and propose solutions to solve them. Previous research also suggests that continuous improvement initiatives must be integrated into organizational culture to be successful [40] and are not something that can be introduced all at once because change should take place gradually and not radically [5]. Imai described kaizen as a state of mind as opposed to a finite task [1]. To be implemented successfully, kaizen should not be seen as an independent activity, but rather as complementary to usual work [41]. We found support for the idea that managers need to dedicate sufficient resources to implementing kaizen—especially in its initial phase—if they want nurses to perceive the approach as part of their work and to participate continuously.

Fifth, policies that weld the team together are fundamental. Combining teamwork training with continuous improvement initiatives may not only enhance process measures, but also improve quality outcomes, such as patient safety [42]. Not all nurses participated in kaizen by contributing their own ideas, and in Hospital B some nurses did not participate at all. Yet we found that nurses who participated regularly tended to enjoy the approach because it facilitated teamwork and promoted team spirit. This result is consistent with the work of Knechtges and Decker [3], who describe teamwork as critical to implementing kaizen successfully, and of Drotz and Poksinska [36], who show that all team members should contribute to make change happen. We also observed that understaffing seemed to impede participation. Additionally, managers who relied on agency staff and units with high staff turnover experienced challenges to keep everybody involved. Managers may therefore want to avoid allowing a heavy workload

to undermine the integration of continuous improvement programs in work routines. This result is in line with previous literature that has shown long-lasting groups to achieve better outcomes [43].

Finally, health care practitioners need to implement kaizen in a structured way. Providing staff members with fixed times and physical space for collaboration is important for implementing continuous improvement techniques successfully [14]. The well-defined responsibilities and processes in Hospital A made the approach clear to the entire team and fostered participation. By holding regular meetings, Hospital A established working routines to convey the message that kaizen is a team-oriented approach that must be performed on a regular basis. In both hospitals, although not equally successful, a number of communication tools integrated kaizen efficiently into the work routine. Dashboards, for example, made change initiatives comprehensible, helped their implementation status to be tracked, and increased compliance with new codes of conduct.

## Limitations and further research

Our study has several important limitations, some of which provide opportunities for further research, which we describe and discuss below.

One limitation of this study is that it included a small number of hospitals. Future efforts should strive to incorporate as many sites as feasible bearing in mind the specific constraints of the research project and context. Another limitation is that the two of the researchers (KS and RB) did not participate in the on-site interviews with the nurses. However, these researchers had the chance to familiarize themselves with the data [44] by reading the transcribed interviews, having a series of discussions with the interviewer (CP), and receiving an introduction to the concept of kaizen and its principles by a team of medical experts (AG, KH, BT). Additionally, to help establish the trustworthiness of our qualitative research methods and results [32, 45], we sought to ensure dependability by providing thorough and transparent documentation of our research interest, methodological choices, and qualitative results in the manuscript and its supporting information [46].

Nevertheless, there are two additional points to bear in mind when considering the transferability of our findings. First, the two acute care hospitals we examined in this study were private and profit-orientated. Although previous research on the hospital sector in Switzerland has shown that there is no significant relationship between (1) profit orientation and hospital ownership and (2) cost-efficiency [19], scholars may nevertheless wish to examine whether these organizational characteristics influence the way hospitals engage with continuous improvement techniques. Furthermore, the hospitals we selected had implemented kaizen with differing degrees of success, which poses the further question of which characteristics are shared by hospitals that are equally successful at implementing kaizen. In this vein, it might be worthwhile in future research to enrich our conclusions by selecting organizations that have implemented the technique in a similar way but at different points in order to explore how the participation of employees evolves over time. Future research may also wish to expand the scope of this study by verifying whether its results hold in settings other than that of inpatient acute care, such as outpatient or long-term care.

Data collection is a second factor that should be considered when interpreting the findings of this study and judging their transferability. While the purposive technique we used to select the interviewees enabled us to gain deep insight into the work environment of both hospitals, it may also have led us to place a disproportionate amount of attention to some experiences the nurses had had with kaizen. For example, there were several interviewees who had started working in their unit fewer than six months before the interview and their observations may

have been influenced by limited knowledge of the work context, the kaizen approach, or both. Yet, purposive sampling helps researchers obtain an in-depth understanding of the phenomenon of interest [30] and is therefore suited to our exploratory approach. When revisiting our work, future researchers could consider using other or additional sampling techniques such as snowball or maximum variation [47], thus aiming to identify key informants who could contribute additional insights and perceptions that would otherwise remain undiscovered. Moreover, according to the social-desirability argument—a common bias occurring in many areas of social sciences that rely on self-reporting values—interviewees may provide answers that do not reflect their real opinions but rather are convenient or socially acceptable [48]. However, we do not expect reporting biases to have distorted our findings substantially because most nurses were not overly shy in criticizing hospital policies or the behaviors of their supervisors. By giving us an intimate look into their working place, the nurses enabled us not only to capture and explore their perceptions of kaizen, but also to realize that this managerial technique —no matter how beneficial it can be in some situations for both employee and organization— is not necessarily a panacea for all problems and aspirations managers may have.

## Conclusion

When implemented successfully, kaizen can reinforce team spirit and increase job satisfaction and commitment among nursing staff in hospitals, enabling the continuous improvement of the organization. To reap these benefits, however, health care managers need to enable nurses to implement the approach in a structured and sustained manner. Drawing upon in-depth qualitative data from diverse examples of implementation, we suggest six strategies for doing so. Health care managers need to (1) show nursing staff how to implement kaizen whenever necessary; (2) endorse each-opinion-counts policies; (3) promulgate the progress achieved in a comprehensive *and* timely manner by showing the entire team how kaizen can improve quality of care; (4) establish an organizational culture that fosters open dialogue across hierarchy levels; (5) ensure team stability and cohesion; and (6) provide employees with infrastructure and communication tools that enable the adoption of ideas.

Employees are among the most important assets of any organization. We believe that the role of employees is even more decisive in non-consumer goods industries like health care because patients depend on the work of caregivers for high-quality treatment and psychological and emotional support. In our view, nursing teams who have more say in everyday decision-making also have greater potential to increase patient satisfaction and quality outcomes. In this regard, kaizen offers health care professionals a practical way to improve the quality of care through small and continuous changes in their workplace.

## Supporting information

**S1 Fig. A graphical illustration of the decision rules we developed to specify the scope of the categories we used in the qualitative content analysis of the summarized interviews.** (PNG)

**S1 Table. The interview guide, which we used for the semi-structured interviews.** (DOCX)

**S2 Table. Overview of primarily and secondary categories, as wells as text markers we used in the content analysis of the interviews.** Organized by interviewee and hospital. (XLSX)

**S3 Table. A minimal underlying data set, which contains illustrative quotes we used to summarize the interviews by site, interviewee, and question.**
(XLSX)

**S4 Table. Overview of the decision rules we developed to specify the scope of the categories we applied to perform the qualitative content analysis of the interviews.**
(DOCX)

## Acknowledgments

The authors would like to thank the nurses for participating in the on-site interviews and to acknowledge the valuable feedback of Tim Brand, Benedikt Englert, and Eva Wild on the manuscript.

## Author Contributions

**Conceptualization:** Kosta Shatrov, Camilla Pessina, Kaspar Huber, Bernhard Thomet, Andreas Gutzeit, Carl Rudolf Blankart.

**Data curation:** Camilla Pessina.

**Formal analysis:** Kosta Shatrov, Camilla Pessina, Carl Rudolf Blankart.

**Investigation:** Kosta Shatrov, Camilla Pessina.

**Methodology:** Kosta Shatrov, Camilla Pessina, Carl Rudolf Blankart.

**Project administration:** Kosta Shatrov, Camilla Pessina, Kaspar Huber, Bernhard Thomet, Andreas Gutzeit, Carl Rudolf Blankart.

**Software:** Kosta Shatrov, Camilla Pessina.

**Supervision:** Andreas Gutzeit, Carl Rudolf Blankart.

**Validation:** Kaspar Huber, Bernhard Thomet, Andreas Gutzeit.

**Visualization:** Kosta Shatrov.

**Writing – original draft:** Kosta Shatrov, Camilla Pessina, Carl Rudolf Blankart.

**Writing – review & editing:** Kosta Shatrov, Camilla Pessina, Kaspar Huber, Bernhard Thomet, Andreas Gutzeit, Carl Rudolf Blankart.

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
