## [Decision Letter · Decision Letter 0]

19 May 2021

PONE-D-21-09173

Improving health care from the bottom up:

Factors for the successful implementation of kaizen in acute care hospitals

PLOS ONE

Dear Dr. Kosta Shatrov,

Thank you for submitting your manuscript to PLOS ONE. After careful consideration, we feel that it has merit but does not fully meet PLOS ONE’s publication criteria as it currently stands. Therefore, we invite you to submit a revised version of the manuscript that addresses the points raised during the review process.

We look forward to receiving your revised manuscript.

Kind regards,

Sharon Mary Brownie

Academic Editor

PLOS ONE

Journal Requirements:

[The author(s) received no specific funding for this work.].   

We note that one or more of the authors are employed by a commercial company: Hirslanden AG

Reviewers' comments:

Reviewer's Responses to Questions

**Comments to the Author**

1. Is the manuscript technically sound, and do the data support the conclusions?

Reviewer #1: Partly

Reviewer #2: Yes

Reviewer #3: Partly

2. Has the statistical analysis been performed appropriately and rigorously? 

Reviewer #1: Yes

Reviewer #2: Yes

Reviewer #3: N/A

3. Have the authors made all data underlying the findings in their manuscript fully available?

Reviewer #1: Yes

Reviewer #2: Yes

Reviewer #3: No

4. Is the manuscript presented in an intelligible fashion and written in standard English?

Reviewer #1: Yes

Reviewer #2: Yes

Reviewer #3: No

5. Review Comments to the Author

Reviewer #1: Kaizen Management Solutions is a market thought in organization consulting, recruiting, training and staffing solutions. The article transplanted the idea and method into the field of nursing management, which has good theoretical and practical significance.BUT there is still some problems need to solve:

Both hospitals you selected were private for-profit acute care hospitals, and this may restrict

the external validity of our findings, Although the article objectively describes the limitation of research in the discussion section, but for readers, they mightly are not quite sure how private, nonprofit hospitals are managed in the Swissland, what difference with the public-run hospitals, and therefore you had better provide a detailed background in the introduction to the article.

It is appropriate to use the qualitative research method for the research of organizational management psychology, but there are some problems in the presentation of the results of the qualitative research, for example, the article only lists the original words of the interviewees, there is no rigorous qualitative interview research method in accordance with the method of subject words, even lack of summarize, distilling, which is very imprecise for the analysis of the results.

There are still many problems in the format and standardization of the article, such as the content of the abstracts, there is a lack of a comprehensive and profound description of the qualitative results, and simply listing the results, therefore, when the conclusion is read, we feels a bit excessive of the results, and partial lack of organic fitness of the conclusion.

Reviewer #2: The authors have successfully provide clear and direct gaps to the study. The discussion and conclusion have delivered and highlighted the relevant gap to fill and strategies to tackle the gaps.

However the reference should be arranged according to alphabetical order to ease the reader to refer to the relevant citation.

Reviewer #3: 1.About studied hospitals, “The two hospitals (hereinafter referred to as Hospital A and Hospital B) had recently implemented kaizen”. As the results show that kaizen does not work well in hospital B, to better understand how kaizen improve health care from the bottom up, why didn’t you choose hospitals that had succeeded in implementing kaizen? Or at different stage of implementing kaizen?

2.This study is from the perspective of nurses, and focus on the results. Are there structural procedures how the hospitals design kaizen? If the authors can add the hospitals’ design of kaizen, readers would understand the whole picture clearly.

3.There are some typos, such as “Organizational culture e was…”

6. PLOS authors have the option to publish the peer review history of their article (what does this mean?). If published, this will include your full peer review and any attached files.

Reviewer #1: **Yes: **Xingming Li

Reviewer #2: No

Reviewer #3: No

---

## [Author Response · Author response to Decision Letter 0]

29 Jun 2021

Revision of our manuscript entitled “Improving health care from the bottom up: Factors for the successful implementation of kaizen in acute care hospitals”

Dear Editor, dear Reviewers, dear Editorial Team,

On behalf of all authors of the paper, I would like to express our gratitude for your helpful comments. We thank all of the reviewers for their thorough reviews and constructive feedback. We have carefully revised the manuscript accordingly and have provided additional data, making our methodology and findings fully transparent to the readership of PLOS ONE. We believe that our manuscript has benefited greatly as a result.

Below please find our detailed point-by-point replies to your comments. 

---

Editor’s comments:

Authors’ response: Thank you for this comment. We have consulted the style templates once again and have adapted our 

manuscript accordingly. In particular, we have revised the formatting of the headings and the tables in the manuscript so that they now adhere to the style requirements set out in the style templates.

Authors’ response: Thank you. To ensure data transparency and address any concerns about the rigor of our methodology, we have provided more data that support our findings and conclusions. In particular, supplementary Table S2 shows the preliminary qualitative content analysis of the interviews with the nurses in both hospitals. Unfortunately, we cannot provide the text of the original interviews to the readers of PLOS ONE because we do not have the informed consent of the interviewed nurses to do so. However, we now provide an extended minimal underlying data set that contains anonymized data that cannot be used to discover the identity of the interviewed persons. The data set can be found in supplementary Table S3. It consists of illustrative quotes from all interviews and can be used to review and reproduce the qualitative content analysis we performed in the study.

In addition, supplementary Table S4 and supplementary Figure S1 show readers the decision rules we used to analyze the illustrative quotes and how we applied them in the second round of our analysis. By applying these decision rules, we achieved an agreement between authors in 90% of the cases. In the final round of our analysis, we solved any disagreement by discussing the quotes in the more global context of the interviews.

[The author(s) received no specific funding for this work.]. 

We note that one or more of the authors are employed by a commercial company: Hirslanden AG

Authors’ response: Thank you for making us aware that we need to amend our Funding Statement. We have revised this statement by complying with the requirements described above, as follows:

“The authors received no specific funding for this work. The authors KH, BT, AG are employees of Hirslanden AG, the owner of the two acute care hospitals investigated in the study. The funder provided support in the form of salaries for authors KH, BT, AG, but did not have any role in the study design, data collection and analysis, decision to publish, or preparation of the manuscript. The specific roles of these authors are articulated in the ‘author contributions’ section.”

Additionally, we have now ensured that specific roles of all authors are articulated in the “author contributions’ section”.

Authors’ response: We have updated the Competing Interests Statement as follows:

“All authors of this manuscript have read the journal's policy and have the following competing interests: KS, CP and RB declare that they have no conflicts of interest. KH, BT, and AG are employed by Hislanden AG, the hospital organization that was examined in this study. The commercial affiliation of the authors KH, BT, and AG does not alter our adherence to PLOS ONE policies on sharing data and materials. Hislanden AG does not profit financially or otherwise from the study results. All authors declare that no other competing interests exist.”

---

Response to Reviewers

Reviewer #1:

Authors’ response: Thank you very much for reading our manuscript carefully and for the positive comments. We feel we have been able to improve our paper substantially as a result. 

R1: Kaizen Management Solutions is a market thought in organization consulting, recruiting, training and staffing solutions. The article transplanted the idea and method into the field of nursing management, which has good theoretical and practical significance. BUT there is still some problems need to solve:

Both hospitals you selected were private for-profit acute care hospitals, and this may restrict

the external validity of our findings, Although the article objectively describes the limitation of research in the discussion section, but for readers, they mightly are not quite sure how private, nonprofit hospitals are managed in the Swissland, what difference with the public-run hospitals, and therefore you had better provide a detailed background in the introduction to the article.

Authors’ response: Thank you for pointing this out. We agree that discussing the validity of our findings in the limitations section was insufficient, so we now provide more details about private hospitals in Switzerland, particularly how they are managed. In doing so, we seek to help readers better understand the context of our study. Following your advice, we have also provided the international readership of PLOS ONE with more information about the Swiss health system, in particular with respect to the ownership status of hospital organizations. Specifically, we have added the following passages:

(1) the Introduction section (page 4, lines, lines 61-64):

“This trend [cost containment] is especially true for countries that refund inpatient care treatments through diagnosis-related groups, a reimbursement system that induces hospitals to focus on cost-efficiency by providing financial incentives to deliver health care at below-average costs.”

As well as (starting on page 6, line 97):

“Inpatient care in Switzerland is reimbursed by health insurers at a flat rate based on diagnosis related groups [19]. This system was introduced in Switzerland in 2012 and applies to all hospitals regardless of their ownership status or profit-orientation. In 2013, about 30% of all acute care hospitals in Switzerland were privately owned, and provided mostly standard surgical and elective treatments [19]. Farsi and Filippini have shown that there are generally no significant differences in cost-efficiency between Swiss hospitals of different ownership status; rather, the cost-driving factors are higher levels of teaching activities in some hospitals and a broader range of specialization in others, and both are associated with lower cost-efficiency [20].

Although the two hospitals that participated in our study were private and profit-oriented, they were comparable to public and non-profit hospitals in terms of a wide range of indicators, including the number of outpatient consultations, number of days in inpatient care, staffing (measured in full-time equivalents), and the overall complexity of treatments provided (measured by a case mix index) [21]. However, there are some important differences between the hospitals examined in this study and the average Swiss hospital, public ones in particular.

The most important difference is that the majority of specialist physicians in the two participating hospitals were not employed by the hospitals but affiliated with them and hired as locums. These physicians provided treatment to patients by renting the infrastructure of the hospital and a team of supporting physicians (anesthetists, emergency and intensive care physicians). However, some of the specialists were directly employed by the hospitals or had post-doctoral degrees and a range of teaching tasks, something that would be more typical of a university hospital. Second, compared to the Swiss median, the two hospitals treated a larger proportion of patients who had a private insurance plan. Such plans enable the patients to choose their treating physician and to stay in a single-bed room. Lastly, although both public and private hospitals have the same range of tasks and obligations in Switzerland, private hospitals are not eligible for public funding in certain domains, such as investments in building infrastructure.

Importantly, the two hospitals are not exclusive private clinics, but an integral part of the Swiss health system in their provision of general acute care. In Switzerland, private hospitals are reimbursed in the same manner as public ones, resulting in financial incentives that are independent of ownership status and profit orientation. The Swiss hospital sector comprises a spectrum ranging from non-profit university hospitals funded by public funds that usually treat the most complex cases to niche private hospitals working almost exclusively with privately insured patients and focusing on selective treatments. Overall, the two hospitals we examined are—irrespective of the important features discussed above—fairly similar to what can be described as the average Swiss hospital.”

(2) The Limitations and further research sub-section of the Discussion section (page 27, lines 527-531):

“Even though previous research on the hospital sector in Switzerland has shown that there is no significant relationship between profit orientation and hospital ownership on the one hand, and cost-efficiency on the other hand [20], scholars of continuous improvement techniques may nevertheless want to examine whether these hospital characteristics limit the validity of our findings.”

R1: It is appropriate to use the qualitative research method for the research of organizational management psychology, but there are some problems in the presentation of the results of the qualitative research, for example, the article only lists the original words of the interviewees, there is no rigorous qualitative interview research method in accordance with the method of subject words, even lack of summarize, distilling, which is very imprecise for the analysis of the results.

Authors’ response: Thank you for making us aware that we should provide our readers with more details about our methods of content analysis. To explain these more thoroughly, and in a more objective and transparent way, we have included the following supporting information in the revised version of the manuscript:

• S2 Table. Overview of primary and secondary categories, as wells as text markers used in the content analysis of the interviews. Organized by interviewee and hospital.

 We exported S2 Table from MAXQDA to show readers the primary and secondary categories we used in the content analysis, and the themes to which we assigned these categories.

• S3 Table. A matrix of illustrative quotes used to summarize the interviews by site, interviewee, and question.

 The content analysis in S3 Table gives the process by which we identified illustrative quotes from all 30 interviews. Please note that the quotes in S3 Table have been anonymized because we do not have informed consent of the nursing teams to disseminate the text of the transcribed interviews. However, researchers willing to trace or reproduce our analysis may use the underlying minimal data set we have now provided as a supplementary digital file (Supplementary Table S3), as well as the information provided in the other supplementary files.

In the next stage of our analysis, we summarized the interviews by exporting the illustrative quotes to the matrix shown in S3 Table. The matrix is organized by hospital, interviewee, and interview question. Then, we applied the Organizational Transformation Model by assigning the illustrative quotes to one of the six primary and 15 secondary categories of the model. Lastly, we have shown some of the illustrative quotes in Table 2 (starting on page 13, at line 248) or in the text to help readers better understand how nurses perceived and implemented kaizen in the two acute care hospitals we examined. We have sought to include quotes from as many nurses as possible and in consideration of the characteristics of our sample including age, gender, tenure, and specialization.

• S4 Table. Overview of the decision rules developed to specify the scope of the categories used in the qualitative content analysis of the summarized interviews.

 This table portrays all categories and sub-categories of the six dimensions of the Organizational Transformation Model we used for the content analysis of illustrative quotes of the summarized interviews (see S3 Table). After applying these decision rules, we achieved agreement in 90% of cases. In the third and final iteration of the content analysis of the illustrative quotes, the two analysts (KS and RB) resolved through discussion any disagreements related to how the quotes had been categorized.

• S1 Figure. Overview of the decision rules developed to specify the scope of the categories used in the qualitative content analysis of the summarized interviews.

 This figure demonstrates the logic we applied in the second iteration of the analysis of the summarized interviews (as shown in S3 Table). After the first iteration round, we achieved an agreement of 68%. Consequently, we developed a set of decision rules to specify the scope of the (sub-)categories of the Organizational Transformation Model. These decision rules are graphically illustrated in S4 Figure and set out in the following S5 Table (see more details below).

R1: There are still many problems in the format and standardization of the article, such as the content of the abstracts, there is a lack of a comprehensive and profound description of the qualitative results, and simply listing the results, therefore, when the conclusion is read, we feels a bit excessive of the results, and partial lack of organic fitness of the conclusion.

Authors’ response: Thank you for raising these important issues. We believe that by addressing your comments, we were able to better demonstrate the rigor of our research. 

We have restructured the abstract in the following way:

(1) We moved the following sentence from the Background section to the Conclusions section of the abstract: “From our findings, we derived six coping strategies to help health care practitioners implement kaizen for the benefit of their organization and employees: […]” (page 3, lines 41-42), thus highlighting better the major contribution of our work, i.e., the coping strategies we derived to help health practitioners to implement kaizen management;

(2) We now state more clearly that we included nurses from different specialties;

(3) We have described our findings in more detail;

(4) In the Conclusions section of the abstract, we have now built on the findings we describe in its Results section, thus making our storyline consistent and more transparent.

Next, by providing more tables and figures (see our response to your previous comment for a full description of the supplementary tables and figures we have provided in the revised version of the manuscript), we describe our content analysis in a more objective and comprehensible way. In particular, the way in which we assigned themes and categories and conducted the initial part of the content analysis is shown in S2 Table. Moreover, we now share the decision rules we used to achieve agreement in the concluding phase of the content analysis in S4 Figure and S5 Table. Additionally, we now explain how we justified our selection of illustrative quotes in the manuscript by adding the following passage to the beginning of the Results section (page 13, lines 245-247):

“Additional illustrative quotes—originating from nurses of different age, gender, tenure, and specialty—that demonstrate our findings have been included in the results section.”

We agree that the previous version of the manuscript was somewhat excessive in the volume of results reports, so we have slightly reduced the number of illustrative quotes provided in Table 2 (starting on page 13, at line 248). We have also reduced the extent of the illustrative quotes provided in the Results section by omitting any words that were not essential for conveying the key message of the cited phrase. Consequently, we believe that the updated version of the manuscript strikes the right balance between backing our findings by showing the relevant passages from the interviews, on the one hand, and producing a text that is not burdened with too much detail, on the other. The omitted illustrative quotes can still be found in S3 Table. This being said, we have also slightly modified the format and the content of Table 2 to improve its readability and consistency, in line with improvements we have made to the rest of the Results section (“H1” was replaced by “Hospital A”, and “H” – by “Hospital B”).

Furthermore, we have restructured the Discussion section of the study—similarly to the Results section—depending on whether the implementation strategies we suggest concern the behavior of individual employees or the way in which a hospital system is organized (page 23, lines 437-440):

“Depending on whether they relate to behavioral patterns that specific team members may need to pursue, or, instead, processes and structures that health professionals might wish to establish, the following strategies are assigned to either the individual or organizational level of acute care hospital systems: […]”

By addressing your comment concerning the format and standardization of the manuscript as described above, we feel that we have been able to show the link between the Results and the Discussion sections in a more direct and traceable way. Additionally, Revision Table 1 (created for the purpose of revising the manuscript and not shown in the manuscript) demonstrates how we have structured the findings, implications, and conclusions of our work originating from the six dimensions of the Organizational Transformation Model: 

Revision Table 1. The link between the constituting part of the manuscript based on the Organization Transformation Model (OTM).

 Level Results section Beginning of Discussion section

(Content of Table 3) Coping strategies (Remainder of the Discussion section) Conclusion section

Dimension of the OTM Individual

 Nurses' perspectives • Link between tenure and willingness to share one's opinion (this results do not represent a factor that affects motivation, but rather the result of these factors and is therefore not shown in Table 3)

 1. Managerial support Same as in the column entitled “Coping strategies”

 Job commitment and satisfaction • Visibility of results 3. Visibility of results 

 Team dynamics and processes • Promoting everyday interactions

• Clearly defined processes and roles

 2. Each opinion counts 

 Organi¬zational Infrastructure availability and adoption

 • Tools, e.g., dashboards 6. Structured implementation 

 Human resources and staffing • Team stability

• Team cohesion 5. Team stability and cohesion 

 Resource allocation and culture • Management and head nurse leadership and support

• Adopting employee suggestions

• Communicating the impact of kaizen activities

• Culture of continuous improvement 1. Managerial support

4. Culture of continuous improvement 

As a result, we think that we now convey the main findings of our work in a more succinct way. To better demonstrate the link between the constituting segments of the manuscript, we have aligned the findings included in the Conclusion section to the structure of the Results and Discussion sections. We believe that in addressing your comment, we have succeeded in improving the readability and comprehensibility of our study.

Lastly, we have revised the Conclusion of the manuscript. As a result, the coping strategies we suggest in the Discussion section are mirrored in the Conclusion in the exact same order. Additionally, we have added the following paragraph to underpin the importance of our research and conclude the manuscript with an issue-expert opinion, summarizing the most important message of our study (page 29, lines 570-576):

“Employees are among the most important assets of any organization. We believe that the role of employees is even more decisive in non-consumer goods industries like health care because patients depend on the work of caregivers for high-quality treatment and psychological and emotional support. In our view, nursing teams who have more say in everyday decision-making also have greater potential to increase patient satisfaction and quality outcomes. In this regard, kaizen offers health care professionals a practical way to improve the quality of care through small and continuous changes in their workplace.”

Reviewer #2:

Authors’ response: Thank you very much for reading our study and for your comment that our work highlights an important research gap and makes suggestions as to how this gap can be addressed.

R2: The authors have successfully provide clear and direct gaps to the study. The discussion and conclusion have delivered and highlighted the relevant gap to fill and strategies to tackle the gaps.

However the reference should be arranged according to alphabetical order to ease the reader to refer to the relevant citation.

Authors’ response: Thank you for your comment. Upon your request, we consulted the submission guidelines on the official website of PLOS ONE once again. To the best of our understanding, PLOS ONE uses the numbered in-text citation method, resulting in references being cited in order of first mention. Additionally, we checked the reference list of some of the most recent publications highlighted on the homepage of the journal. The reference lists of these journals are, similarly to our reference list, numbered and ordered by the sequence of appearance of the references in text. Moreover, our reference list comprises all available Digital Object Identifiers (DOI) of studies cited in our manuscript. We believe the DOI will help readers find and refer to the documents we have cited in our study. 

Thanks to your comment that induced us to revise the reference list, we realized that there were some minor issues with one of the references cited. We have now corrected the citation as follows (page 30, lines 598-600):

“Glew DJ, O’Leary-Kelly AM, Griffin RW, van Fleet DD. Participation in Organizations: A Preview of the Issues and Proposed Framework for Future Analysis. J Manage. 1995; 21:395–421. doi: 10.1177/014920639502100302.”

We have also adjusted the corresponding text of the manuscript accordingly (pages 4-5, lines 69-71):

“By expanding the role of less senior employees, whether formally or informally, managers can reinforce employee participation in decision making [7].”

Reviewing the complete reference list once again, we noticed that the scholars of continuous improvement techniques have devoted a considerable amount of attention to the United Kingdom in recent years, albeit not to the same extent as to the United States. Therefore, we have revised the definition of the research gap as follows (pages 5-6, lines 93-95):

“[…] Lastly, by offering evidence from outside the widely studied cases of the United States and the United Kingdom [11,12], we aimed to support mutual learning among countries with different health systems and traditions of nursing education and practice.”

As well as (page 22, lines 412-414):

“Our findings provide insights from a setting outside of the United States or the United Kingdom, which has been the almost exclusive focus of previous research in this area.”

Reviewer #3:

Authors’ response: Thank you very much for reading our work carefully and making us aware of issues we needed to address to better explain our motivation and results to readers. We think that by taking account of your comments in our revision, we have been able to improve the paper substantially.

R3: 1.About studied hospitals, “The two hospitals (hereinafter referred to as Hospital A and Hospital B) had recently implemented kaizen”. As the results show that kaizen does not work well in hospital B, to better understand how kaizen improve health care from the bottom up, why didn’t you choose hospitals that had succeeded in implementing kaizen? Or at different stage of implementing kaizen?

Authors’ response: Thank you for pointing this out to us. We agree with you that the kaizen continuous improvement technique was not implemented equally well in both hospitals. The literature review of D'Andreamatteo et al. (2015) shows that even though continuous improvement techniques have been widely applied in health care organization in the recent decades and can help health professionals improve care provision, the existing literature on continuous improvement management focuses on cases of successful implementation, largely ignoring cases in which techniques such as kaizen have been implemented in a less successful manner. The authors therefore urge scholars to expand the horizon of state-of-the-art literature, by including cases of less successful implementation. D'Andreamatteo et al. (2015) and Edmondson (2004) advocate learning from failure, because contrasting cases of more and less successful implementation make clear how specific measures can be implemented in a more efficient way.

We included hospitals that were at a very similar stage of implementing kaizen because we expected we would otherwise experience difficulties discriminating between effects of the way in which kaizen had been implemented and time lag effects. By selecting two hospitals that were fairly comparable—not only in terms of implementation progress, but also in terms of size, specialization, and organization—we expected that any time lag effects would be negligible. As a result, we argue that any differences in the results achieved using kaizen are likely to be attributable to the way the technique was implemented. Based on this assumption, we sought to capture the differences between the two hospitals and make suggestions about how health professionals should implement the technique if they seek to improve health care provision. We conclude that stable and coherent nursing teams, managerial support, and kaizen tools such as dashboards may facilitate the implementation of the technique.

However, our approach has some limitations, and we elaborate on these in the Discussion section. In the new version of the manuscript, we have included the following sentence in the Limitations sub-section of the Discussion section (page 27, lines 531-534):

“Another factor conceivably restricting the external validity of this study is the selection of the hospitals. For that reason, our preliminary findings can be tested by examining more hospitals that have either succeeded at implementing kaizen or are at a different stage of kaizen implementation.”

R3: 2.This study is from the perspective of nurses, and focus on the results. Are there structural procedures how the hospitals design kaizen? If the authors can add the hospitals’ design of kaizen, readers would understand the whole picture clearly.

Authors’ response: We are thankful that you have raised this important question. To address it, we added the following information to the Data collection sub-section of the Methodology section (pages 9-10, starting at line 167):

“In Hospital A, an implementation working group, whose main objective was to set out the conditions for the nursing teams to be able to start contributing ideas, was appointed by the management team of the hospital at the beginning of the implementation phase. To communicate the concept of kaizen to the staff in all units, the working groups defined a set of general and specific aims. While the general aims involved encouraging nurses to seek to identify problems in the workplace and subsequently solve them, the specific ones defined concrete measures intending to facilitate collaboration. Such specific measures included the advice that kaizen meetings should be held on a regular basis and that nurses should aspire to be role models to their colleagues from other units by working with and for each other. Lastly, the process of implementing kaizen was represented graphically in the recreation rooms of Hospital A and defined in five steps as follows: (1) identify waste; (2) make an improvement suggestion; (3) prioritize the suggestion and define action to implement it; (4) take the defined actions; (5) measure the success of the actions.

The initial phase of implementing kaizen was defined in a similar way in Hospital B. Together, the quality management team and the nursing team of the hospital elaborated a strategy to design the way in which kaizen would be implemented in all inpatient care units. The strategy of Hospital B sought to encourage employees to question existing working procedures, inform their colleagues in case action is required, and eventually make a collective effort to improve their workplace. In addition, responsible persons received special training and subsequently served as an important reference point to the rest of team.”

Second, we have revised the following paragraph of the Results section to include more information about the design of kaizen (pages 16-17, starting at line 295): 

“Both hospitals implemented kaizen in a similar way. Regular discussion rounds were introduced—up to once in a fortnight in Hospital A and on a weekly basis in Hospital B—and kaizen dashboards were installed in all units. The dashboards represented whiteboards, so nurses were able to make improvement suggestions and track their status by documenting their ideas using markers and post-it notes. The dashboards had been placed in easily accessible places such as break rooms and kitchens. Nevertheless, there were also some differences. Hospital A started implementing kaizen in 2010; Hospital B did so in 2011—first in all inpatient departments, and two years later in the intensive care unit. Although the dashboards in both hospitals were divided into sections that were devoted to the tasks of contributing ideas and defining a set of actions for their implementation, Hospital B did not use a pie diagram to visualize the implementation status of ideas. Hospital A offered compulsory introductory training to all nurses. In Hospital B, however, the quality management team visited units to answer any questions nurses had during the initial phase of implementation. Hospital A set the goal of implementing 20 ideas in each unit per year, whereas Hospital B aimed to have 36 meetings dedicated to kaizen during the first year of implementation. Within this year, Hospital A implemented 958 suggestions in total, whereas Hospital B implemented 321 suggestions.”

Lastly, some of the illustrative quotes we included in the article show how nurses experienced the implementation of kaizen. Here are some examples:

(1) Table 2 (starting on page 13, at line 248):

““We have a kaizen training, in which the system is explained and applied. I think this is good for employees who are new and have no previous experience with the system.” (Hospital A/07)”

(2) Table 2 (starting on page 13, at line 248):

“The team meeting gives us structure […] At the same time, it’s hard to make a meaningful contribution if you desperately need to do something or you’re quite busy.” (Hospital B/21)

(3) Page 16, lines 284-287:

“‘Once a month, the kaizen [meeting] takes place and the employees meet in the office. [...] The problems are then discussed within the team, and we see what can be improved, what the options are, and which person or people are responsible for implementing it’ (Hospital A/12)’”

R3: 3.There are some typos, such as “Organizational culture e was…”

Authors’ response: Thank you for pointing these out to us. We have addressed these.

We look forward to receiving your reply. Thank you very much for your time and consideration.

Yours faithfully,

The corresponding author on behalf of all authors of the paper

---

## [Decision Letter · Decision Letter 1]

2 Aug 2021

PONE-D-21-09173R1

Improving health care from the bottom up:

Factors for the successful implementation of kaizen in acute care hospitals

PLOS ONE

Dear Dr. Kosta Shatrov,

Thank you for submitting your manuscript to PLOS ONE. After careful consideration, we feel that it has merit but does not fully meet PLOS ONE’s publication criteria as it currently stands. Therefore, we invite you to submit a revised version of the manuscript that addresses the points raised during the review process.

We look forward to receiving your revised manuscript.

Kind regards,

Sharon Mary Brownie

Academic Editor

PLOS ONE

Journal Requirements:

Reviewers' comments:

Reviewer's Responses to Questions

**Comments to the Author**

Reviewer #3: All comments have been addressed

Reviewer #4: (No Response)

2. Is the manuscript technically sound, and do the data support the conclusions?

Reviewer #3: Yes

Reviewer #4: Yes

3. Has the statistical analysis been performed appropriately and rigorously? 

Reviewer #3: Yes

Reviewer #4: N/A

4. Have the authors made all data underlying the findings in their manuscript fully available?

Reviewer #3: Yes

Reviewer #4: Yes

5. Is the manuscript presented in an intelligible fashion and written in standard English?

Reviewer #3: Yes

Reviewer #4: Yes

6. Review Comments to the Author

Reviewer #3: (No Response)

Reviewer #4: The authors have sufficiently addressed the reviewers’ comments. The manuscript is well-written and articulate. The authors have provided sufficient information including additional supplementary tables that have enriched the methods and findings of the manuscript. However, the limitation and further research section sounds like discussion of limitation for a quantitative rather than a qualitative research. The authors should revise the section with a view of clearly highlighting the trustworthiness of the study.

7. PLOS authors have the option to publish the peer review history of their article (what does this mean?). If published, this will include your full peer review and any attached files.

Reviewer #3: No

Reviewer #4: No

---

## [Author Response · Author response to Decision Letter 1]

20 Aug 2021

Rebuttal letter accompanying the second revision of our manuscript entitled “Improving health care from the bottom up: Factors for the successful implementation of kaizen in acute care hospitals”

Dear Editor, dear Reviewers, dear Editorial Team,

On behalf of all authors of the paper, I would like to thank you for providing your thoughtful comments on the revised version of our paper. We have carefully revised the manuscript accordingly, reorienting the section labeled “Limitations and further research” to the context of our qualitative study. We believe that our manuscript has benefited significantly as a result.

Below please find our detailed point-by-point replies to your comments. 

Authors’ response: Thank you for this comment. None of the references we cite in the paper has been retracted. After reviewing the complete reference list, we made a few minor changes mentioned in R2 Table 1. All changes have been highlighted in red on a grey background. The “Identifier Revision 1” (ID_R1) stands for the reference number during the first revision round, whereas the second identifier (ID_R2) stands for the updated reference numbers (during this revision round):

R2 Table 1. An overview of all changes to the reference list.

ID_R1 ID_R2 Author(s) Note Revised reference (if needed)

1 1 Imai M. Kaizen No changes needed 

2 2 Bhuiyan N, Baghel A. No changes needed 

3 3 Knechtges P, Decker MC No changes needed 

4 4 Mazzocato P, Stenfors-Hayes T, von Thiele Schwarz U, Hasson H, Nyström ME. No changes needed 

5 5 Vera A, Kuntz L No changes needed 

6 6 Bailey C, Madden A, Alfes K, Fletcher L. No changes needed 

7 7 Glew DJ, O’Leary-Kelly AM, Griffin RW, van Fleet DD No changes needed 

8 8 Radnor Z, Boaden R (1) Title corrected (by adding a question mark) as suggested on the publisher's website: https://www.tandfonline.com/doi/abs/10.1111/j.1467-9302.2008.00610.x?journalCode=rpmm20 (accessed August 11, 2021);

(2) We have found a DOI, but did not add it to the reference list, because it was not active: DOI: 10.1111/j.1467-9302.2008.00610.x Radnor Z, Boaden R. Lean in Public Services—Panacea or Paradox? Public Money & Management. 2008; 28:3–7.

9 9 Comtois J, Paris Y, Poder TG, Chaussé S. Page range was incomplete Comtois, Jonathan, et al. « L'approche Kaizen au Centre hospitalier universitaire de Sherbrooke (CHUS) : un avantage organisationnel significatif », Santé Publique, vol. 25, 2013, pp. 169-77.

10 10 Iannettoni MD, Lynch WR, Parekh KR, McLaughlin KA No changes needed 

11 11 D'Andreamatteo A, Ianni L, Lega F, Sargiacomo M No changes needed 

12 12 Filser LD, da Silva FF, Oliveira OJ de Author's name corrected ("de Oliveira" instead of "Oliveira") Filser LD, da Silva FF, de Oliveira OJ de. State of research and future research tendencies in lean healthcare: a bibliometric analysis. Scientometrics. 2017; 112:799–816. doi: 10.1007/s11192-017-2409-8.

13 13 Jacobson GH, McCoin NS, Lescallette R, Russ S, Slovis CM No changes needed 

14 14 New S, Hadi M, Pickering S, Robertson E, Morgan L, Griffin D, et al. No changes needed 

15 15 Edmondson AC No changes needed 

16 16 Young T, Brailsford S, Connell C, Davies R, Harper P, Klein JH No changes needed 

17 17 Jørgensen F, Boer H, Gertsen F No changes needed 

18 18 De Pietro C, Camenzind P, Sturny I, Crivelli L, Edwards-Garavoglia S, Spranger A, et al. We have adopted the citation suggested by the World Health Organization (cf. https://apps.who.int/iris/handle/10665/330252; accessed August 11, 2021) World Health Organization. Regional Office for Europe, European Observatory on Health Systems and Policies, De Pietro, Carlo, Camenzind, Paul, Sturny, Isabelle. et al. (‎2015)‎. Switzerland: health system review. World Health Organization. Regional Office for Europe. https://apps.who.int/iris/handle/10665/330252.

19 19 Farsi M, Filippini M. No changes needed 

20 20 Hauser T, Salgado-Thalmann E, Bachmann M We have added the editor (i.e., Federal Office of Public Health), as well as the follwoing access link: https://spitalstatistik.bagapps.ch/data/download/kzp18_publication.pdf?v=1592291836 Hauser T, Salgado-Thalmann E, Bachmann M. Health insurance statistics. Key indicators of the Swiss hospitals 2018. Federal Office of Public Health 2020 [cited 6 May 2021]. Available from: https://spitalstatistik.bagapps.ch/data/download/kzp18_publication.pdf?v=1592291836.

21 21 Lukas CV, Holmes SK, Cohen AB, Restuccia J, Cramer IE, Shwartz M, et al. No changes needed 

22 22 Corley KG, Gioia DA No changes needed 

23 23 Bansal P, Corley K No changes needed 

24 24 Becker TE, Billings RS, Eveleth DM, Gilbert NL No changes needed 

25 25 Fedor DB, Caldwell S, Herold DM No changes needed 

26 26 Meyer JP, Allen NJ No changes needed 

27 27 Scott‐Ladd B, Travaglione A, Marshall V No changes needed 

28 28 Gill P, Stewart K, Treasure E, Chadwick B No changes needed 

29 29 Hirslanden We have now added the place of publication of the hospital statistics report provided by the Hirslanden hospitals. Hirslanden. Hospital Statistics. Fiscal Year 1.4.2017-31.3.2018. Glattpark, Switzerland; 2018.

30 30 Patton MQ No changes needed 

31 31 Sullivan JL, Adjognon OL, Engle RL, Shin MH, Afable MK, Rudin W, et al. No changes needed 

N/A 32 Lincoln YS, Guba EG Added during the revision of the limitations section 

32 33 Herzberg F, Mausner B, Snyderman BB. No changes needed 

33 34 Bassett‐Jones N, Lloyd GC Title corrected (by adding a question mark) Bassett‐Jones N, Lloyd GC. Does Herzberg's motivation theory have staying power? Journal of Mgmt Development. 2005; 24:929–43. doi: 10.1108/02621710510627064.

34 35 Wong CA, Spence Laschinger HK, Cummings GG No changes needed 

35 36 Drotz E, Poksinska B No changes needed 

36 37 Collar RM, Shuman AG, Feiner S, McGonegal AK, Heidel N, Duck M, et al. No changes needed 

37 38 Zaheer S, Ginsburg L, Chuang Y-T, Grace SL No changes needed 

38 39 Gagne M, Koestner R, Zuckerman M (1) Author's name corrected ("é" instead of "e");

(2) Title corrected (by leaving out the superfluous character "1") Gagneeé M, Koestner R, Zuckerman M. Facilitating Acceptance of Organizational Change: The Importance of Self-Determination1. J Appl Social Pyschol. 2000; 30:1843–52. doi: 10.1111/j.1559-1816.2000.tb02471.x

39 40 Clark DM, Silvester K, Knowles S No changes needed 

40 41 Simon RW, Canacari EG No changes needed 

41 42 Robertson E, Morgan L, New S, Pickering S, Hadi M, Collins G, et al. No changes needed 

42 43 Brännmark M, Holden RJ No changes needed 

N/A 44 Braun V, Clarke V

 Added during the revision of the limitations section 

N/A 45 Nowell LS, Norris JM, White DE, Moules NJ. Added during the revision of the limitations section 

N/A 46 Tobin GA, Begley CM Added during the revision of the limitations section 

43 47 Palinkas LA, Horwitz SM, Green CA, Wisdom JP, Duan N, Hoagwood K No changes needed. 

44 48 Fisher RJ, Katz JE No changes needed 

45 N/A Zuckerman M Removed during the revision of the limitations section 

46 N/A Diefenbach T. Removed during the revision of the limitations section 

47 N/A Hunter DJ Removed during the revision of the limitations section 

48 N/A Radnor ZJ, Holweg M, Waring J Removed during the revision of the limitations section 

Response to Reviewers

Reviewer #4: The authors have sufficiently addressed the reviewers’ comments. The manuscript is well-written and articulate. The authors have provided sufficient information including additional supplementary tables that have enriched the methods and findings of the manuscript. However, the limitation and further research section sounds like discussion of limitation for a quantitative rather than a qualitative research. The authors should revise the section with a view of clearly highlighting the trustworthiness of the study.

Authors’ response: Thank you for making us aware of this. We have carefully revised the section as follows:

1. We specify how we aimed to keep a self-critical account of our research by describing the extent to which our study meets the trustworthiness criteria suggested by Lincoln and Guba (1985) and illustrated by Nowell, et al. (2017). In so doing, we now highlight the rigor of our study, providing researchers with the opportunity to evaluate the trustworthiness of the research process (pages 27-28, lines 534-538):

“[…] to help establish the trustworthiness of our qualitative research methods and results [32,45], we sought to ensure dependability by providing thorough and transparent documentation of our research interest, methodological choices, and qualitative results in the manuscript and its supporting information [46].”

Additionally, we have now demonstrated that we have adhered to Braun and Clarke’s (2006) recommendations for familiarizing oneself with the qualitative data (page 27, lines 531-534):

“[…] these researchers [KS and RB] had the chance to familiarize themselves with the data [44] by reading the transcribed interviews, having a series of discussions with the interviewer (CP), and receiving an introduction to the concept of kaizen and its principles by a team of medical experts (AG, KH, BT).”

2. Another aspect of our revision of the limitations section is that we consolidated the limitations related to rigor and trustworthiness—and to transferability in particular—into two major groups of concern. The first group relates to how we chose to select the hospitals (page 28, lines 540-541):

“First, the two acute care hospitals we examined in this study were private and profit-orientated. […]”

The second group of limitations is related to the sampling technique we used to conduct the on-site interviews (page 28, lines 554-557):

“While the purposive technique we used to select the interviewees enabled us to gain deep insight into the work environment of both hospitals, it may also have led us to place a disproportionate amount of attention to some experiences the nurses had had with kaizen. […]”

By providing these detailed descriptions, we aim to help readers who seek to transfer our findings to another context to judge the transferability of our findings (Lincoln and Guba, 1985).

3. We now use language that is more positive. For example (page 28, lines 553-554):

“Data collection is a second factor that should be considered when interpreting the findings of this study and judging their transferability.”

4. We also streamlined the section to enhance its readability and relevance by deleting the following passages: 

“[…] our analysis relied on self-reported, retrospective, and cross-sectional data, which may have decreased the reliability of our results. […] It is also conceivable that some respondents may have tended to take credit for successes while denying responsibility for failure, leading to so-called self-serving bias [45].”

 “Indeed, interviewees not being a reliable source of information—because of unconscious biases or conscious attempts to mislead the interviewer—is a common methodological problem in qualitative empirical research [46].”

 “In addition, health care workers are more likely to talk openly about problems if their work unit and organization have psychologically safe work climates [15]. Thus, ironically, higher-performing units may look worse based on self-reported data simply because interviewees were more willing to speak their mind.“ 

 “Lastly, and more generally, questioning the idea of implementing kaizen in health care was not the object of this study, although translating managerial concepts from industry into complex service industries like health care may not be an appropriate endeavor in the first place [49], especially at the systems level [50].”

5. To demonstrate in a more convincing way that the limitations discussed do not pose a major limitation to our results we highlight the credibility of our research by adding the following passage (page 29, lines 567-573):

“However, we do not expect reporting biases to have distorted our findings substantially because most nurses were not overly shy in criticizing hospital policies or the behaviors of their supervisors. By giving us an intimate look into their working place, the nurses enabled us not only to capture and explore their perceptions of kaizen, but also to realize that this managerial technique—no matter how beneficial it can be in some situations for both employee and organization—is not necessarily a panacea for all problems and aspirations managers may have.“

6. In order to highlight the trustworthiness of our approach, we added the following sentence to the Methods section (page 11, lines 210-212): 

“In the next step, to enhance the credibility of analysis [32], two researchers (KS and RB) used the OTM framework to analyze the interviews.”

Minor editorial changes:

Finally, we made two changes to the Results sections to remove the following minor inconsistencies:

1. It should be “Hospital B”, not “Hospital 2” in (page 20, lines 378-379):

“At the beginning it [kaizen] has an effect – I would say for about […] 4 weeks, or even only for 10 days, and then a lot, not everything, but a lot is forgotten’ (Hospital B/28)”

2. We completed this sentence by adding the authors’ names (page 20, lines 385-386):

“Noting that the study of kaizen has focused so far on success stories, D'Andreamatteo, et al. [11] and Filser, et al. [12] advocate learning from examples of less effective implementation.”

We look forward to receiving your reply. Thank you very much for your time and consideration.

Yours faithfully,

The corresponding author on behalf of all authors of the paper

---

## [Decision Letter · Decision Letter 2]

1 Sep 2021

Improving health care from the bottom up:

Factors for the successful implementation of kaizen in acute care hospitals

PONE-D-21-09173R2

Dear Dr. Kosta Shatrov,

We’re pleased to inform you that your manuscript has been judged scientifically suitable for publication and will be formally accepted for publication once it meets all outstanding technical requirements.

Kind regards,

Sharon Mary Brownie

Academic Editor

PLOS ONE

Reviewers' comments:

Reviewer's Responses to Questions

**Comments to the Author**

1. If the authors have adequately addressed your comments raised in a previous round of review and you feel that this manuscript is now acceptable for publication, you may indicate that here to bypass the “Comments to the Author” section, enter your conflict of interest statement in the “Confidential to Editor” section, and submit your "Accept" recommendation.

Reviewer #4: All comments have been addressed

2. Is the manuscript technically sound, and do the data support the conclusions?

Reviewer #4: (No Response)

3. Has the statistical analysis been performed appropriately and rigorously? 

Reviewer #4: (No Response)

4. Have the authors made all data underlying the findings in their manuscript fully available?

Reviewer #4: (No Response)

5. Is the manuscript presented in an intelligible fashion and written in standard English?

Reviewer #4: (No Response)

6. Review Comments to the Author

Reviewer #4: (No Response)

7. PLOS authors have the option to publish the peer review history of their article (what does this mean?). If published, this will include your full peer review and any attached files.

Reviewer #4: No

---

## [Editor Report · Acceptance letter]

3 Sep 2021

PONE-D-21-09173R2 

Improving health care from the bottom up:
Factors for the successful implementation of kaizen in acute care hospitals 

Dear Dr. Shatrov:

I'm pleased to inform you that your manuscript has been deemed suitable for publication in PLOS ONE. Congratulations! Your manuscript is now with our production department. 

Kind regards, 

on behalf of

Professor Sharon Mary Brownie 

Academic Editor

PLOS ONE